# Bivariate Causal Discovery for Categorical Data via Classification with Optimal Label Permutation

**Yang Ni**
Department of Statistics
Texas A&M University
College Station, TX 77843
yni@stat.tamu.edu

## Abstract

Causal discovery for quantitative data has been extensively studied but less is known for categorical data. We propose a novel causal model for categorical data based on a new classification model, termed classification with optimal label permutation (COLP). By design, COLP is a parsimonious classifier, which gives rise to a provably identifiable causal model. A simple learning algorithm via comparing likelihood functions of causal and anti-causal models suffices to learn the causal direction. Through experiments with synthetic and real data, we demonstrate the favorable performance of the proposed COLP-based causal model compared to state-of-the-art methods. We also make available an accompanying R package `COLP`, which contains the proposed causal discovery algorithm and a benchmark dataset of categorical cause-effect pairs.

## 1   Introduction

Discovering causality from observational data has seen rapid development in recent years partly because knowledge of causality is desired in many areas where controlled experimentation is very difficult, infeasible, or expensive to carry out. Particularly, for continuous and count data, numerous methods and theories have been developed [Shimizu et al., 2006, Hoyer et al., 2009, Zhang and Hyvärinen, 2009, Mooij et al., 2010, Janzing et al., 2012, Chen et al., 2014, Sgouritsa et al., 2015, Hernandez-Lobato et al., 2016, Marx and Vreeken, 2017, Blöbaum et al., 2018, Park and Park, 2019, Choi et al., 2020, Tagasovska et al., 2020]. All of these methods, in essence, exploit the quantitative nature of continuous and count data in discovering causality. Therefore, they are not applicable to categorical data for which the values can only be interpreted qualitatively. For example, while $Y = g(X) + E$ may be a reasonable causal model for continuous data, the interpretation of such model for categorical data, although possible [Peters et al., 2010, Suzuki et al., 2014] under certain circumstances, is much less natural because the order of magnitude of the values of categorical data is arbitrary and meaningless [Cai et al., 2018].

In general, causal discovery for categorical data is much less studied. What is known to date is that the causal model $X \to Y$ can be identified if $X$ and $Y$ are ordinal [Ni and Mallick, 2022], if $X$ admits a hidden compact representation $Y' = f(X)$ such that $|Y'| < |X|$ ($|\cdot|$ denotes the cardinality) and $X \to Y' \to Y$ [Cai et al., 2018, Qiao et al., 2021], if the exogenous variable $E$ of the structural causal model $Y = f(X, E)$ has entropy that does not scale with the number of categories [Compton et al., 2020, 2022], if $P(X)$ and $P(Y|X)$ are independent random variables [Liu and Chan, 2016], or if the categorical variables $X$ and $Y$ are binary and they do not share the same marginal distribution [Wei et al., 2018]. Note that causal discovery methods that focus on identifying Markov equivalence classes [Spirtes et al., 2000, Lam et al., 2022] are not directly applicable to bivariate causal discovery

36th Conference on Neural Information Processing Systems (NeurIPS 2022).

problems as $X \to Y$ and $Y \to X$ are Markov equivalent. However, when additional variables are available, they may be able to identify direct or indirect causal relationship between $X$ and $Y$.

In this paper, we propose a novel causal model for categorical data based on classification with optimal label permutation (COLP). COLP itself is a new classifier, which is more parsimonious than multinomial regression. COLP is inspired by ordinal regression, which has considerably lower model complexity than multinomial regression. Unfortunately, by design, ordinal regression is only applicable to categorical responses that admit a natural ordering, e.g., human satisfaction (low, medium, and high). However, many categorical variables (e.g., choice of sports from {gymnastics, boxing, volleyball}) do not appear to have natural orderings but we argue that for the purposes of prediction, the response $Y$ may be ordered in a meaningful way depending on the predictor $X$. For instance, if one wants to to predict a person's choice of sports $Y \in$ {gymnastics, boxing, volleyball} based on his/her height $X$, it would make sense to order gymnastics $<$ boxing $<$ volleyball because on average volleyball players are taller than boxers who in turn tend to be taller than gymnasts. On the other hand, if the prediction of $Y$ is based on the person's strength, another ordering, volleyball $<$ gymnastics $<$ boxer, may be more suitable. In either case, once the ordering has been figured out, an ordinal regression can be applied to model and predict $Y$ given $X$. Of course, determining the ordering of $Y$ could be subjective and tedious. The proposed COLP model is precisely designed to automatically find the best category ordering in an objective way. As expected, its model complexity is between multinomial regression and ordinal regression.

The main objective of this paper is causal discovery for categorical data. It turns out that the parsimony of COLP is quite useful in that regard – while causal models based on multinomial regression are non-identifiable, we prove that the proposed COLP-based causal models are identifiable under the causal Markov and causal sufficient assumptions. Our experiments with synthetic and real data show that the proposed method outperforms state-of-the-art alternative methods.

## 2 Proposed Method

We first introduce the classification model COLP in Section 2.1. COLP may be of interest by itself as a new classifier but the focus of this paper is to build a causal model based on COLP, which is presented in Section 2.2.

### 2.1 Classification with Optimal Label Permutation

Let $Y \in \{1, \dots, L\}$ be a categorical response variable with $L$ levels and let $\boldsymbol{X} = (X_1, \dots, X_S)^T$ be a $S$-dimensional predictor vector. Note that if $L = 2$, ordinal logistic regression, nominal logistic regression, and the proposed COLP are equivalent and hence hereafter we always assume $L > 2$. Later, $\boldsymbol{X}$ will be dummy variables representing a categorical predictor with $S$ levels but, for now, we present COLP for a general set of predictors.

If $Y$ is ordered, an ordinal regression is often used,

$$P(Y \leq \ell | \boldsymbol{X}) = F(\gamma_\ell - \boldsymbol{X}^T \boldsymbol{\beta}), \ \ \ell = 1, \dots, L, \tag{1}$$

where $F$ is some link function (e.g., standard normal or logistic CDF), $\gamma_1 < \dots < \gamma_L$ are a set of thresholds, and $\boldsymbol{\beta} \in \mathbb{R}^S$ are ordinal regression coefficients. Equation (1) implies the conditional probability distribution $P(Y = \ell | \boldsymbol{X}) = F(\gamma_\ell - \boldsymbol{X}^T \boldsymbol{\beta}) - F(\gamma_{\ell-1} - \boldsymbol{X}^T \boldsymbol{\beta})$ for $\ell \in \{1, \dots, L\}$ where $\gamma_0 = -\infty$, $\gamma_1 = 0$ (for parameter identifiability), and $\gamma_L = \infty$. Therefore, effectively, the model complexity (i.e., the number of parameters) of an ordinal regression is $L - 2 + S$.

If $Y$ is nominal with no natural ordering, a multinomial (logistic) regression can be used instead,

$$P(Y = \ell | \boldsymbol{X}) = \frac{e^{\boldsymbol{X}^T \boldsymbol{\beta}_\ell}}{\sum_{\ell'=1}^{L} e^{\boldsymbol{X}^T \boldsymbol{\beta}_{\ell'}}}, \ \ \ell = 1, \dots, L, \tag{2}$$

where $\boldsymbol{\beta}_\ell$ are category-specific regression coefficients and, for parameter identifiability, $\boldsymbol{\beta}_L = \boldsymbol{0}$. The effective model complexity is $(L - 1) \times S$, which is strictly greater than the model complexity of an ordinal regression, $L - 2 + S$ for $L > 2$ and $S > 1$. Even though multinomial regression is obviously also applicable to ordinal data by simply ignoring the ordering, ordinal regression is often preferred over multinomial regression in this case because of parsimony.

Here, we propose a new classification model, which is more parsimonious than multinomial regression and is useful beyond ordinal categorical data. The general idea is to introduce a permutation $\sigma : \{1, \ldots, L\} \mapsto \{1, \ldots, L\}$, which orders the categories so that the ordinal regression is applicable. Specifically, we propose the following probability model,

$$P(Y \leq \ell | \boldsymbol{X}) = F(\gamma_{\sigma(\ell)} - \boldsymbol{X}^T \boldsymbol{\beta}), \ \ell = 1, \ldots, L, \tag{3}$$

which is similar to ordinal regression (1) but with an important additional parameter $\sigma \in \Sigma$ where $\Sigma$ is the collection of all permutations of size $L$. Because any ordering $\sigma$ and its reverse $\widetilde{\sigma}$ (i.e., $\sigma(i) < \sigma(j)$ if and only if $\widetilde{\sigma}(i) > \widetilde{\sigma}(j)$) would lead to equivalent ordinal regression models, for parameter identifiability, we assume $\sigma(1) < \sigma(2)$. Therefore, the effective size of $\sigma$ is $L - 2$ because once $\sigma(3), \ldots, \sigma(L)$ are fixed, $\sigma(1)$ and $\sigma(2)$ are fixed due to the contraint. Consequently, the overall complexity of the proposed COLP model is $L - 2 + S + L - 2 = 2L + S - 4$, which is less than the complexity of a multinomial regression, $(L - 1) \times S$, for $L, S > 2$. Similarly to the ordinal regression, (3) implies the conditional probability mass function,

$$P(Y = \ell | \boldsymbol{X}) = F(\gamma_{\sigma(\ell)} - \boldsymbol{X}^T \boldsymbol{\beta}) - F(\gamma_{\sigma(\ell)-1} - \boldsymbol{X}^T \boldsymbol{\beta}), \ \ell = 1, \ldots, L. \tag{4}$$

As we mentioned in Section 1, although a categorical variable may not have a natural ordering, for the purpose of modeling and prediction, they may be ordered in a meaningful way depending on the predictors. The proposed COLP, by including ordering as a parameter, can automatically find the best ordering in an objective manner. In addition, even for categorical variables that have natural orderings, the proposed COLP may still be preferred over both ordinal regression and multinomial regression. For instance, in one of later real data examples, $Y =$ shelf placement $\in \{1, 2, 3\}$ (counting from the floor) and $X =$ cereal manufacturer. To predict $Y$ based on $X$, it makes more sense to use a less natural ordering $1 < 3 < 2$ for $Y$ as shoppers are more likely to buy products on the middle shelf than either the top or bottom. In fact, when we ran COLP on this data, $1 < 3 < 2$ was identified as the optimal ordering. Moreover, COLP and the multinomial regression had the same goodness of fit, which was better than that of the ordinal regression. COLP had the best out-of-sample prediction, followed by the ordinal regression, and the multinomial was the worst. For this example, COLP had the right model complexity to achieve the best model fit as well as the best out-of-sample prediction.

## 2.2 COLP-Based Causal Discovery

Next, we build a causal model based on COLP. Let $Y \in \{1, \ldots, L\}$ and $X \in \{1, \ldots, S\}$ with $L, S > 2$. The COLP-based causal model considers two competing causal hypotheses,

$$M_0 : X \rightarrow Y \ \text{vs} \ M_1 : Y \rightarrow X$$

with (observational) probability mass functions,

$$P_{X \rightarrow Y}(X = s, Y = \ell) = P_{X \rightarrow Y}(X = s) P_{X \rightarrow Y}(Y = \ell | X = s),$$
$$P_{Y \rightarrow X}(X = s, Y = \ell) = P_{Y \rightarrow X}(Y = \ell) P_{Y \rightarrow X}(X = s | Y = \ell),$$

where $P_{X \rightarrow Y}(X = s)$ and $P_{Y \rightarrow X}(Y = \ell)$ are multinomial with probabilities $\boldsymbol{\omega} = (\omega_1, \ldots, \omega_S)$ and $\boldsymbol{\rho} = (\rho_1, \ldots, \rho_L)$, and $P_{X \rightarrow Y}(Y = \ell | X = s)$ and $P_{Y \rightarrow X}(X = s | Y = \ell)$ take similar forms as (4),

$$P_{X \rightarrow Y}(Y = \ell | X = s) = F(\gamma_{\sigma(\ell)} - \boldsymbol{X}^T \boldsymbol{\beta}) - F(\gamma_{\sigma(\ell)-1} - \boldsymbol{X}^T \boldsymbol{\beta}),$$
$$P_{Y \rightarrow X}(X = s | Y = \ell) = F(\eta_{\pi(s)} - \boldsymbol{Y}^T \boldsymbol{\alpha}) - F(\eta_{\pi(s)-1} - \boldsymbol{Y}^T \boldsymbol{\alpha}),$$

where $\boldsymbol{X} \in \{0, 1\}^S$ and $\boldsymbol{Y} \in \{0, 1\}^L$ are dummy variable representation of $X$ and $Y$, and $\sigma \in \Sigma$ and $\pi \in \Pi$ are permutations of $\{1, \ldots, L\}$ and $\{1, \ldots, S\}$. In summary, causal model $M_0 : X \rightarrow Y$ is parameterized by $(\boldsymbol{\omega}, \boldsymbol{\beta}, \boldsymbol{\gamma}, \sigma)$ with $\boldsymbol{\gamma} = (\gamma_2, \ldots, \gamma_{L-1})$ and $\boldsymbol{\beta} \in \mathbb{R}^S$ whereas $M_1 : Y \rightarrow X$ is parameterized by $(\boldsymbol{\rho}, \boldsymbol{\alpha}, \boldsymbol{\eta}, \pi)$ with $\boldsymbol{\eta} = (\eta_2, \ldots, \eta_{S-1})$ and $\boldsymbol{\alpha} \in \mathbb{R}^L$.

Like regression, the proposed COLP-based casual model (complexity $= 2L + 2S - 5$) is more parsimonious than a saturated bivariate multinomial model (complexity $= S \times L - 1$). In fact, a multinomial causal model where $P_{X \rightarrow Y}(Y = \ell | X = s)$ is multinomial regression has the same complexity as the saturated model. Therefore, a multinomial causal model is essentially just a reparameterization of a joint multinomial distribution, which of course can be factorized in both causal and anti-causal directions, and hence is not identifiable. Now, the question is: can the parsimonious COLP-based casual model break the symmetry? The answer is yes, which will be formally established in the next section.

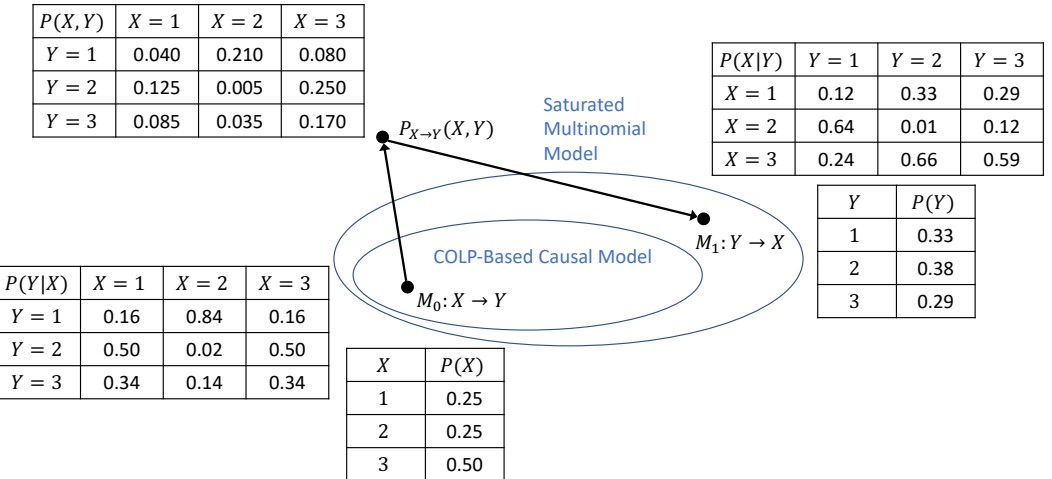

| $P(X,Y)$ | $X=1$ | $X=2$ | $X=3$ |
|---|---|---|---|
| $Y=1$ | 0.040 | 0.210 | 0.080 |
| $Y=2$ | 0.125 | 0.005 | 0.250 |
| $Y=3$ | 0.085 | 0.035 | 0.170 |

| $P(X\|Y)$ | $Y=1$ | $Y=2$ | $Y=3$ |
|---|---|---|---|
| $X=1$ | 0.12 | 0.33 | 0.29 |
| $X=2$ | 0.64 | 0.01 | 0.12 |
| $X=3$ | 0.24 | 0.66 | 0.59 |

| $Y$ | $P(Y)$ |
|---|---|
| 1 | 0.33 |
| 2 | 0.38 |
| 3 | 0.29 |

| $P(Y\|X)$ | $X=1$ | $X=2$ | $X=3$ |
|---|---|---|---|
| $Y=1$ | 0.16 | 0.84 | 0.16 |
| $Y=2$ | 0.50 | 0.02 | 0.50 |
| $Y=3$ | 0.34 | 0.14 | 0.34 |

| $X$ | $P(X)$ |
|---|---|
| 1 | 0.25 |
| 2 | 0.25 |
| 3 | 0.50 |

Figure 1: Illustration of causal identifiability of COLP-based causal model. The set of joint distributions $P(X,Y)$ that can be represented by the COLP-based causal model is a subset of those represented by the saturated multinomial model (this relation is indicated by ellipses). A specific COLP-based causal model $M_0 : X \to Y$ is given by $\boldsymbol{\omega} = (0.25, 0.25, 0.5), \gamma = 1, \boldsymbol{\beta} = (1, -1, 1)^T$, $\sigma(1) = 1, \sigma(2) = 3$, and $\sigma(3) = 2$. These parameter values determine the conditional probability $P(Y|X)$ and the marginal probability $P(X)$, which in turn define the joint probability $P(X,Y)$. Although it is easy to find $P(X|Y)$ and $P(Y)$ for the anti-causal model $M_1 : Y \to X$ from the joint probability $P(X,Y)$, $M_1$ is no longer in the class of COLP-based causal models. Hence, if causal models are constrained to be COLP-based, the correct causal direction $X \to Y$ can be identified.

## 2.3 Identifiability

Before stating our main identifiability theorem, we first provide intuition as to why multinomial regression-based causal models are non-identifiable whereas the proposed COLP-based causal models are identifiable. As mentioned in Section 2.2, multinomial regression-based causal models are simply reparameterization of a saturated bivariate multinomial model whereas COLP-based causal models are more parsimonious. We represent such relation as a Venn diagram in Figure 1. For a given COLP-based causal model (represented by the dot in the inner ellipse), say $M_0 : X \to Y$ with $X, Y \in \{1, 2, 3\}$, its conditional probability $P(Y|X)$ and marginal probability $P(X)$ (represented by the probability tables at the bottom left corner) are determined by its specific parameter values, say $\boldsymbol{\omega} = (0.25, 0.25, 0.5), \gamma = 1, \boldsymbol{\beta} = (1, -1, 1)^T, \sigma(1) = 1, \sigma(2) = 3$, and $\sigma(3) = 2$. The conditional and marginal probability distributions define the joint distribution $P(X,Y)$ represented by the dot and the probability table at the top left corner. Now consider a causal model with a reversed direction $M_1 : Y \to X$. Since the joint distribution $P(X,Y)$ can always factorize into $P(Y)$ and $P(X|Y)$ (represented by the probability tables at the top right corner), it is obvious that $M_0 \equiv M_1$ under such factorization. But $M_1$, represented by the dot in the outer ellipse, does not belong to the class of COLP-based causal models anymore. In summary, when constrained to COLP-based causal models, this particular example of $M_0$ does not have an equivalent model. The identifiability theorem below shows that this is true in general.

**Theorem 1** *If there is no unmeasured confounder, the link function $F(\cdot)$ is a fixed real analytic function[1], and $F'(\cdot)$ is nowhere zero, then for almost all $(\boldsymbol{\omega}, \boldsymbol{\beta}, \gamma, \sigma)$, there does not exist $(\boldsymbol{\rho}, \boldsymbol{\alpha}, \boldsymbol{\eta}, \pi)$ such that $M_0 \equiv M_1$, i.e., $P_{X \to Y}(X = s, Y = \ell) = P_{Y \to X}(X = s, Y = \ell)$ for all $(s, l) \in \{1, \ldots, S\} \times \{1, \ldots, L\}$.*

All proofs are provided in the Supplementary Material. No unmeasured confounder is a common assumption in prior causal discovery work for categorical data [Peters et al., 2010, Suzuki et al., 2014, Liu and Chan, 2016, Cai et al., 2018, Compton et al., 2020]. The requirements on the link function $F(\cdot)$ are quite mild; well-known link functions such as probit and logistic satisfy them.

---

[1]A real function is said to be analytic if it is infinitely differentiable and matches its Taylor series in a neighborhood of every point.

---

**Algorithm 1** Greedy Search: MLE of COLP

---

**Input:** data $(x_1, y_1), \ldots, (x_n, y_n)$, initial parameters $\boldsymbol{\omega}, \boldsymbol{\beta}, \boldsymbol{\gamma}, \sigma$
Compute $M(\sigma) = \max_{\boldsymbol{\omega}, \boldsymbol{\beta}, \boldsymbol{\gamma}} \prod_{i=1}^{n} P_{X \to Y}(X = x_i, Y = y_i | \boldsymbol{\omega}, \boldsymbol{\beta}, \boldsymbol{\gamma}, \sigma)$
Set $M_\star = M(\sigma)$
**repeat**
  Initialize $Improvement = false$
  **for** all permutation $\sigma'$ reachable from $\sigma$ **do**
    Compute $M(\sigma')$
    **if** $M(\sigma') > M_\star$ **then**
      Set $\sigma = \sigma'$ and $M_\star = M(\sigma')$
      Set $Improvement = true$
    **end if**
  **end for**
**until** $Improvement$ is $false$
**Output:** maximized likelihood $M_\star$

---

Next, we show that asymptotically we can correctly identify the true causal model.

**Theorem 2** *If $M_0 : X \to Y$ is the true data generating model, the likelihood of $M_0$ is asymptotically greater than that of the anti-causal model $M_1 : Y \to X$.*

Theorems 1 and 2 suggest a simple causal discovery algorithm based on maximum likelihood estimation (MLE). For a dataset with $n$ observations, $(x_1, y_1), \ldots, (x_n, y_n)$, we conclude $M_0 : X \to Y$ if

$$\max_{\boldsymbol{\omega}, \boldsymbol{\beta}, \boldsymbol{\gamma}, \sigma} \prod_{i=1}^{n} P_{X \to Y}(X = x_i, Y = y_i | \boldsymbol{\omega}, \boldsymbol{\beta}, \boldsymbol{\gamma}, \sigma) > \max_{\boldsymbol{\rho}, \boldsymbol{\alpha}, \boldsymbol{\eta}, \pi} \prod_{i=1}^{n} P_{Y \to X}(X = x_i, Y = y_i | \boldsymbol{\rho}, \boldsymbol{\alpha}, \boldsymbol{\eta}, \pi),$$

and conclude $M_1 : Y \to X$ otherwise. The MLE can be carried out in two steps. In the first step, for every $\sigma \in \Sigma$, we maximize the likelihood over $\boldsymbol{\omega}, \boldsymbol{\beta}, \boldsymbol{\gamma}$ through the standard MLE of ordinal regression by treating $\sigma(y_1), \ldots, \sigma(y_n)$ as ordered labels, $M(\sigma) = \max_{\boldsymbol{\omega}, \boldsymbol{\beta}, \boldsymbol{\gamma}} \prod_{i=1}^{n} P_{X \to Y}(X = x_i, Y = y_i | \boldsymbol{\omega}, \boldsymbol{\beta}, \boldsymbol{\gamma}, \sigma)$. Then in the second step, we pick the largest $M(\sigma)$ among all $\sigma \in \Sigma$. This exhaustive search over all permutations is feasible when the number of categories is small. For categorical data with a moderately large number of categories, an iterative greedy search algorithm (Algorithm 1) can be used instead. At each iteration, we compute the MLE of ordinal regression for all the permutations that can be reached from the current permutation by switching the order of two elements. We replace the current permutation by the permutation with the largest increase in likelihood and stop the algorithm when the likelihood can no longer be improved.

## 3 Experiments

### 3.1 Synthetic Data

We first assessed the performance of the proposed COLP-based causal discovery method with three sets of synthetic data. For comparison, we considered a recent categorical discovery method based on hidden compact representation (HCR, [Cai et al., 2018]).

#### 3.1.1 Scenario 1: Small Number of Categories

We generated data with the number of categories $L = S = 5$ and varying sample size $n = 50, 100, \ldots, 1000$. The true parameters were set as $\boldsymbol{\omega} = (1/5, 1/5, 1/5, 1/5, 1/5)$, $\boldsymbol{\beta} \sim N(0, \boldsymbol{I}_5)$, $\sigma(\ell) = \ell, \forall \ell$, and $\boldsymbol{\gamma}$ chosen to have balanced class size for each variable. Both the exhaustive (COLP-Exhaustive) and greedy (COLP-Greedy) versions of the COLP-based causal discovery algorithm were applied. The results based on 500 repeat simulations are summarized in Figure 2a. COLP-Exhaustive and COLP-Greedy had virtually the same accuracy in identifying the correct causal directions, both of which increased with the sample size, which empirically verified Theorems 1 and 2, and uniformly outperformed HCR. We also computed the Kendall's Tau between the estimated category ordering and the true ordering. Kendall's Tau close to 1 indicates a good estimation. The average Kendall's

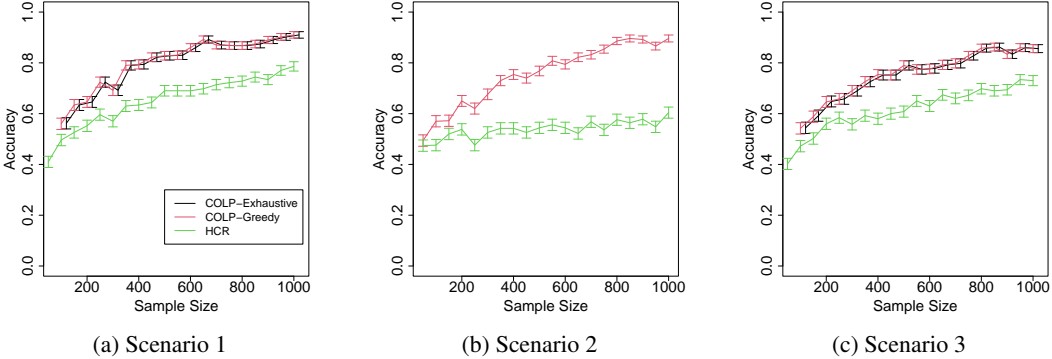

(a) Scenario 1         (b) Scenario 2         (c) Scenario 3

Figure 2: Synthetic Data. Average accuracy of causal identification for COLP-Exhausitve, COLP-Greedy, and HCR across different sample sizes and scenarios based on 500 repeat simulations. Standard errors are represented by the error bars. The accuracy curves of COLP-Exhaustive in (a) and (c) are slightly shifted to the right for visualization.

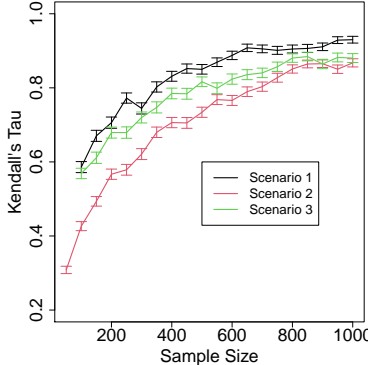

Figure 3: Synthetic Data. Average Kendall's Tau of ordering estimation for COLP-Greedy, across different sample sizes and scenarios based on 500 repeat simulations. Standard errors are represented by the error bars.

Tau of COLP-Greedy is reported in Figure 3. As sample size increased, the ordering estimation improved as expected.

In the Supplementary Material, we present two additional results under this scenario: (i) we performed an ablation study to demonstrate the importance of learning the category ordering, and (ii) we investigated how estimation of causal direction and label permutation vary as the number of categories increases.

### 3.1.2 Scenario 2: Larger Number of Categories

We now increased the number of categories to $L = S = 10$ while keeping all the other simulation parameters the same. We did not apply COLP-Exhaustive in this scenario. As shown in Figure 2b, COLP-Greedy outperformed HCR across all sample sizes and the margins were wider than those in Scenario 1. The ordering estimation had the similar increasing trend in Kendall's Tau as sample size increased as in Scenario 1 (Figure 3).

### 3.1.3 Scenario 3: Hidden Confounders

While our identifiability theory assumes no unmeasured confounders, we empirically tested the sensitivity of our method to the presence of confounders. We generated trivariate categorical data $(X, Y, Z)$ from the following true causal graph with all the simulation parameters kept the same as in Scenario 1,

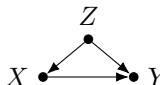

We applied COLP and HCR to $(X, Y)$ only (i.e., $Z$ became a hidden confounder). As shown in Figures 2c and 3, COLP had the best performance and the estimation of causal directions and category orderings approached perfect recovery as sample size increased even in the presence of confounders.

## 3.2 Real Data

We further evaluated the proposed COLP-based causal discovery method with four sets of public real categorical data: (i) Pittsburgh Bridges dataset, (ii) Abalone dataset, (iii) Tübingen Cause-Effect Pairs, and (iv) a newly-created Categorical Cause-Effect Pairs. For comparison, we considered HCR as before and an additional competing method, entropic causal inference (ECI, Compton et al. [2020]). For variables with more than six categories, only the greedy search was applied for COLP-based causal discovery. For variables with fewer categories, both exhaustive and greedy algorithms were applied, which generally produced the same results; therefore, we do not differentiate between the two implementations when reporting the results below for simplicity.

### 3.2.1 Pittsburgh Bridges Dataset

This dataset [Reich and Fenves, 1989] is available from the UCI Machine Learning Repository and was used in previous causal discovery work [Cai et al., 2018]. It has 108 observations and the following 4 true cause-effect pairs: Erected (Crafts, Emerging, Mature, Modern) → Span (Short, Medium, Long), Material (Steel, Iron, Wood) → Span (Short, Medium, Long), Material → Lanes (1, 2, 4, 6), and Purpose (Aqueduct, Highway, Rail, Walk) → Type (Wood, Suspen, Simple-T, Arch, Cantilev, CONT-T). In addition to HCR and ECI, we also applied Markov equivalence class-based causal discovery algorithm, GRaSP [Lam et al., 2022], to all the variables.

The results are presented in Table 1. COLP was able to correctly identify all 4 cause-effect pairs whereas HCR missed 1 pair, ECI missed 2 pairs, and GRaSP correctly identified two direct causal links and two indirect causal links (i.e., directed paths with the correct directions). The effect variables of the first three pairs, Span and Lanes, have natural orderings, namely, Short $<$ Medium $<$ Long and $1 < 2 < 4 < 6$. The optimal orderings identified by COLP perfectly matched them (note that COLP does not take the natural ordering as an input). The effect variable, Type, of the last pair does not have an obvious natural ordering. The optimal ordering was estimated to be Simple-T $<$ Cantilev $<$ CONT-T $<$ Arch $<$ Wood $<$ Suspen and the COLP regression coefficients under $X \to Y$ were estimated to be $\hat{\beta}_{\text{Aqueduct}} = 2.90, \hat{\beta}_{\text{Highway}} = 1.03, \hat{\beta}_{\text{Rail}} = -1.63$, and $\hat{\beta}_{\text{Walk}} = 13.66$. This ordering seems sensible considering that the predictor/cause was Purpose. For example, {Simple-T, Cantilev, CONT-T} bridges are more likely to be used for rail roads whereas {Arch, Wood, Suspen} bridges are more likely to be used for walking. Therefore, their ordering is consistent with the signs of $\hat{\beta}_{\text{Rail}}$ (negative) and $\hat{\beta}_{\text{Walk}}$ (positive).

Table 1: Pittsburgh Bridges Dataset. Correctly (incorrectly) identified causal direction is marked by ✓(✗). For GRaSP, ◯ means a directed path was identified.

| Cause (X) | Effect (Y) | COLP | HCR | ECI | GRaSP |
|-----------|-----------|------|-----|-----|-------|
| Erected | Span | ✓ | ✓ | ✗ | ◯ |
| Material | Span | ✓ | ✓ | ✓ | ✓ |
| Material | Lanes | ✓ | ✓ | ✗ | ◯ |
| Purpose | Type | ✓ | ✗ | ✓ | ✓ |

### 3.2.2 Abalone Dataset

This dataset [Nash et al., 1994] is available from the UCI Machine Learning Repository and was used in previous research [Cai et al., 2018]. It has 4177 observations and the following 3 true cause-effect pairs: Sex (male, female, infant) → Length, Sex → Diameter, and Sex → Height. We discretized Length, Diameter, and Height into 5 categories at their 20%,40%,60%, and 80% quantiles. As in Section 3.2.1, we compared COLP with HCR, ECI, and GRaSP.

The results are reported in Table 2. COLP and ECI were able to correctly identify all 3 cause-effect pairs whereas HCR missed 1 pair, and GRaSP correctly identified one direct causal relationship and one indirect causal relationship, and failed to determine the causal direction of one pair. Because all the effect variables were obtained by discretization at quantiles, they had natural orderings. Again, in all cases, the optimal orderings identified by COLP perfectly matched them.

Table 2: Abalone Dataset. Correctly (incorrectly) identified causal direction is marked by ✓(✗). For GRaSP, ◯ means a directed path was identified.

| Cause (X) | Effect (Y) | COLP | HCR | ECI | GRaSP |
|---|---|---|---|---|---|
| Sex | Length | ✓ | ✓ | ✓ | ◯ |
| | Diameter | ✓ | ✓ | ✓ | ✗ |
| | Height | ✓ | ✗ | ✓ | ✓ |

### 3.2.3 Tübingen Cause-Effect Pairs

This is a well-known causal benchmark dataset [Mooij et al., 2016] (version: 12/20/2017). We picked pair 52, 53, 54, 55, and 105 for testing, which were rarely used in prior work because at least one of the variables in each pair is multivariate. We applied K-means to each multivariate variable with $K = 5$ and used the cluster labels as a categorical variable, and discretized each univariate variable at 5 evenly spaced quantiles.

The results are shown in Table 3. COLP was able to correctly identify all 5 cause-effect pairs whereas HCR missed 1 pair and ECI missed 2 pairs. The effect variables of Pairs 53 and 105 have natural orderings, which matched the optimal orderings identified by COLP.

Table 3: Tübingen Cause-Effect Pairs. Correctly (incorrectly) identified causal direction is marked by ✓(✗).

| Pair | Cause (X) | Effect (Y) | COLP | HCR | ECI |
|---|---|---|---|---|---|
| 52 | $\begin{pmatrix} \text{air temperature} \\ \text{pressure at surface} \\ \text{sea level pressure} \\ \text{relative humidity} \end{pmatrix}$ at day 50 | $\begin{pmatrix} \text{air temperature} \\ \text{pressure at surface} \\ \text{sea level pressure} \\ \text{relative humidity} \end{pmatrix}$ at day 51 | ✓ | ✓ | ✓ |
| 53 | $\begin{pmatrix} \text{wind speed} \\ \text{global radiation} \\ \text{temperature} \end{pmatrix}$ | ozone concentration | ✓ | ✓ | ✗ |
| 54 | $\begin{pmatrix} \text{displacement} \\ \text{horsepower} \\ \text{weight} \end{pmatrix}$ | $\begin{pmatrix} \text{mpg} \\ \text{acceleration} \end{pmatrix}$ | ✓ | ✓ | ✓ |
| 55 | temperature at 16 locations | ozone concentration at 16 locations | ✓ | ✗ | ✓ |
| 105 | grey values of 9 pixels | light intensity | ✓ | ✓ | ✗ |

### 3.2.4 Categorical Cause-Effect Pairs

The Tübingen Cause-Effect Pairs data are largely continuous and may not be the best benchmarks for categorical causal discovery. Hence, we created a categorical causal discovery benchmark dataset using a similar approach as in Mooij et al. [2016]. Specifically, we searched for appropriate datasets in R packages `MASS` and `datasets` for which the pairwise causal relationships should be obvious from the context (e.g., treatment assignment causes treatment effect), and at least one of the variables in each pair is categorical. For non-categorical variable, we discretized it at 5 evenly spaced quantiles. The resulting dataset contains 33 categorical cause-effect pairs and is available in the R package `COLP`.

The results are shown in Table 4. Overall, COLP, HCR, and ECI were able to correctly identify 70%, 61%, and 52% causal-effect pairs, respectively. In terms of the ordering estimation, some results were interesting. For example, as mentioned in Section 2.1, for the "MASS::UScereal" data, the ordering of $Y = $ shelf placement $\in \{1, 2, 3\}$ was estimated to be $1 < 3 < 2$, which matches the fact that middle shelf is the most popular, followed by the top shelf, and the bottom shelf is the

least popular. In fact, under the correct causal direction $X \rightarrow Y$, COLP was better than ordinal and multinomial regressions in terms of both goodness of fit (via within-sample prediction) and out-of-sample prediction (via leave-one-out cross-validation).

Table 4: Categorical Cause-Effect Pairs. Correctly (incorrectly) identified causal direction is marked by ✓(✗).

| Source | Data | Cause (X) | Effect (Y) | COLP | HCR | ECI |
|--------|------|-----------|------------|------|-----|-----|
| MASS | anorexia | Treat | Prewt-Postwt | ✓ | ✗ | ✓ |
| MASS | painters | School | Composition | ✗ | ✓ | ✗ |
| MASS | painters | School | Drawing | ✗ | ✓ | ✓ |
| MASS | painters | School | Colour | ✓ | ✓ | ✗ |
| MASS | painters | School | Expression | ✓ | ✓ | ✗ |
| MASS | birthwt | race | low | ✓ | ✗ | ✗ |
| MASS | bacteria | trt | y | ✓ | ✓ | ✗ |
| MASS | survey | Sex | Clap | ✓ | ✗ | ✓ |
| MASS | survey | Sex | Fold | ✓ | ✗ | ✓ |
| MASS | oats | B | Y | ✓ | ✓ | ✗ |
| MASS | oats | V | Y | ✓ | ✓ | ✓ |
| MASS | oats | N | Y | ✗ | ✓ | ✓ |
| MASS | crabs | sp*sex | FL | ✗ | ✗ | ✗ |
| MASS | crabs | sp*sex | RW | ✓ | ✗ | ✓ |
| MASS | crabs | sp*sex | CL | ✓ | ✗ | ✓ |
| MASS | crabs | sp*sex | CW | ✓ | ✗ | ✓ |
| MASS | crabs | sp*sex | BD | ✓ | ✓ | ✓ |
| MASS | fgl | type | RI | ✗ | ✓ | ✓ |
| MASS | immer | Var | Y1 | ✓ | ✓ | ✗ |
| MASS | immer | Var | Y2 | ✓ | ✗ | ✓ |
| MASS | immer | Loc | Y1 | ✓ | ✓ | ✗ |
| MASS | immer | Loc | Y2 | ✗ | ✗ | ✗ |
| MASS | minn38 | sex | phs | ✗ | ✗ | ✓ |
| MASS | minn38 | fol | hs | ✓ | ✓ | ✗ |
| MASS | minn38 | fol | phs | ✗ | ✓ | ✗ |
| MASS | UScereal | mfr | shelf | ✓ | ✓ | ✗ |
| MASS | UScereal | mfr | vitamins | ✓ | ✓ | ✗ |
| datasets | chickwts | feed | weight | ✗ | ✓ | ✗ |
| datasets | InsectSprays | spray | count | ✓ | ✓ | ✓ |
| datasets | npk | N*P*K | yield | ✓ | ✓ | ✓ |
| datasets | PlantGrowth | group | weight | ✗ | ✗ | ✗ |
| datasets | ToothGrowth | supp*dose | len | ✓ | ✓ | ✓ |
| datasets | warpbreaks | wool*tension | breaks | ✓ | ✗ | ✓ |

## 4   Conclusion

There are a few limitations of the current work. First, our identifiability theory assumes no un-measured confounders. Although our empirical studies suggested that the proposed method was relatively robust to the presence of confounders, it would be interesting to theoretically investigate the identifiability under this scenario. Second, we have focused on bivariate causal discovery. Extending it to multivariate cases would broaden the applicability of the proposed method. Third, the categorical cause-effect pairs dataset can be expanded by surveying more publicly available data.

## Acknowledgments and Disclosure of Funding

This research was partially supported by NSF DMS-2112943, NSF DMS-1918851, and NIH 1R01GM148974-01. We appreciate the constructive comments from anonymous reviewers as well as Spencer Compton and Murat Kocaoglu, which helped improve the paper.

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
