# OpenReview forum: "Bivariate Causal Discovery for Categorical Data via Classification with Optimal Label Permutation"
_NeurIPS.cc/2022/Conference — NeurIPS 2022 Accept_

### Official Review · Reviewer_q4Dr · 2022-07-06

**Rating:** 5
**Confidence:** 3
**Soundness:** 1 poor
**Presentation:** 2 fair
**Contribution:** 2 fair

**Summary:**

This paper proposes a parametric way for causal discovery over categorical data, which is far less studied than continuous data. The proposed method is named "classification with optimal label permutation (COLP)". COLP is inspired by ordinal regression. While the latter only applies to variables with a natural ordering, COLP can extend to categorical variables without a natural ordering - in the sense of prediction response. The corresponding causal model finds an optimal permutation and COLP parameters to satisfy one causal direction, while the other anti-causal direction cannot be satisfied by any COLP model - as is illustrated in the identifiability proof. Technically, the paper uses simple MLE to estimate the optimal COLP model. Experiments with synthetic and real data demonstrate COLP's better performance than SOTAs.

**Questions:**

**1. What are the hidden assumptions?**

It's known that [1] for multinomial distribution or linear Gaussian SEMs, the causal graph can be identified up to its Markov equivalence class (intrinsically on data; no matter which method is used). Then, in this paper, for the causal direction between pairwise categorical data to be identified, there must be parametric assumptions on the data generating process. For example, in the person's choice of sports case,
to predict $Y\in$ {gymnastics, boxing, volleyball} based on one's strength or height $X$, actually here it relies on the fact that $X$ is ordinal categorical, or even continuous. Then,
  - Could the authors please give a mathematical formulation of all the assumptions used in COLP?
  - Are those assumptions stronger or weaker, compared to the existing work, e.g., ordinal regression and multinomial regression?
  - Overall, are those assumptions testifiable from data (e.g., LiNGAM assumption is testifiable by checking scatter plots)?
  - Specifically on the $X\rightarrow Y$ case above with continuous $X$, can COLP be viewed as (or compared with) methods for mixed-type data (e.g., using degenerated/conditional Gaussian score).
  - The meaningful order of $Y$ replies on $X$ (as a regressor), i.e., yields a dependence relationship. Then, if we go beyond the bivariate case to the multivariate case, will this also yield a dependence graph? How to find a root (i.e., with natural ordering) variable? How to ensure that there is no loop in the dependence graph?

**2. The identifiability is poorly explained, and the results seem to be wrong (to me).**

Let's start with the illustrative example in Figure 1. Authors indicate that a COLP-based causal model $M_0:X\rightarrow Y$ is satisfied with specific parameters $\omega = (0.25, 0.25, 0.5)$, $\gamma = 1$, $\beta=(1,-1,1)^\intercal$, and $\sigma([1,2,3])=[1,3,2]$. By the equation in line 106, we have the link function satisfying $F(-1)=0.16, F(0)=0.34, F(1)=0.84, F(2)=0.98$. Then,
 - Seems that the unpermuted $Y$ ($\sigma([1,2,3])=[1,2,3]$) also satisfies a COLP model (with other parameters $\omega, \gamma, \beta$ unchanged): just let $F$ satisfy $F(-1)=0.16, F(0)=0.66, F(1)=0.84, F(2)=0.86$.
 - In general, how does COLP deal with the non-unique permutations for COLP models as above?
 - The paper states that "$M_1:Y\rightarrow X$ is no longer in the class of COLP-based causal models". But it seems that this is incorrect.
   + Suppose we have $\omega=(0.33,0.38,0.29)$, $\gamma=1$, $\beta=(-a,-b,-c)^\intercal$.
   + Yes there does not exist a COLP model for unpermuted $X$: just consider the first two columns ($Y=1,2$), we cannot have a CDF link function $F$ with $F(a)=0.12, F(a+1)=0.76, F(b)=0.33, F(b+1)=0.34$, since $F$ is monotonically increasing.
   + But, e.g., for the permutation of $X$: $\sigma([1,2,3])=[1,3,2]$, there exists $F$ s.t., $F(a)=0.12, F(a+1)=0.36, F(b)=0.33, F(b+1)=0.99, F(c)=0.29, F(c+1)=0.88$ -- i.e., there is a COLP model also for $M_1:Y\rightarrow X$.
 - I tried some examples though. E.g, this is a joint table where $X\rightarrow Y$ can follow a COLP model, but not $Y\rightarrow X$ (verify in a similar way as above):
| P(X,Y) |  X=1  |  X=2  |  X=3  |
|:------:|:-----:|:-----:|:-----:|
|   Y=1  | 0.147 | 0.081 | 0.110 |
|   Y=2  | 0.144 | 0.097 | 0.096 |
|   Y=3  | 0.080 | 0.092 | 0.152 |

 - And this is a joint table where neither $X\rightarrow Y$ nor $Y\rightarrow X$ can be represented by a COLP model:
| P(X,Y) |  X=1  |  X=2  |  X=3  |
|:------:|:-----:|:-----:|:-----:|
|   Y=1  | 0.128 | 0.073 | 0.087 |
|   Y=2  | 0.061 | 0.054 | 0.180 |
|   Y=3  | 0.069 | 0.170 | 0.178 |

 - Overall I think, to check whether there exists a COLP model, it's to check for the conditional probability table, whether there exists a row permutation s.t., for the cumulated summed table along the row axis, the column-wise argsort indices are same for all rows (so that there exists a valid CDF monotonically increasing function to satisfy the same length (of threshold $\gamma$ intervals)). See the code below:
```python
def is_X_to_Y(XY_joint_table):
    '''
    :param XY_joint_table:
        e.g. Figure 1 in the paper,
        +--------+-------+-------+-------+
        | P(X,Y) |  X=1  |  X=2  |  X=3  |
        +--------+-------+-------+-------+
        |   Y=1  | 0.040 | 0.210 | 0.080 |
        +--------+-------+-------+-------+
        |   Y=2  | 0.125 | 0.005 | 0.250 |
        +--------+-------+-------+-------+
        |   Y=3  | 0.085 | 0.035 | 0.170 |
        +--------+-------+-------+-------+
        where axis0 is Y and axis1 is X.
    :return:
        True if there exists a COLP model X->Y, otherwise False.
    '''
    P_Y_given_X = XY_joint_table / XY_joint_table.sum(axis=0)[None]
    for row_perm in itertools.permutations(np.arange(P_Y_given_X.shape[0])):
        permed_P_Y_given_X = P_Y_given_X[list(row_perm)]
        permed_CDF_Y_given_X = np.cumsum(permed_P_Y_given_X, axis=0)
        col_sorted_ids = np.argsort(permed_CDF_Y_given_X[:-1],axis=1)
        if (col_sorted_ids == col_sorted_ids[0]).all():
            return True
    return False
```

 - Then by the code above, I sampled 10000 random joint probability tables with both $X,Y$ of cardinality 3 (I just choose $\alpha=1$ for the Dirichlet distribution without further considerations):
```python
both_direction_cnt, one_direction_cnt, no_direction_cnt = 0, 0, 0
for _ in range(10000):
       joint = np.random.dirichlet(np.ones(9), size=1).reshape((3, 3))
       X_to_Y, Y_to_X = is_X_to_Y(joint), is_X_to_Y(joint.T)
       if X_to_Y and Y_to_X: both_direction_cnt += 1
       elif X_to_Y or Y_to_X: one_direction_cnt += 1
       else: no_direction_cnt += 1
print(both_direction_cnt, one_direction_cnt, no_direction_cnt)
```
The result is that: `7129` are unidentifiable (both directions can have a COLP model); `1059` are unidentifiable (both directions do not have a COLP model), and only `1812` are identifiable (one direction satisfies, and one does not).
 - And with the cardinalities going up, less can be satisfied by COLP model (as expected). E.g, for $5x5=25$ table, the counts are respectively `both_direction_cnt=16, one_direction_cnt=405, no_direction_cnt=9579`.
- This definitely violates the proof of zero measurements (Theorem 1). So, did I understand correctly? Please correct me if anything is wrong.

**3. Questions about experimental results on real data**
 - Instead of accuracy metric as of in simulation results, for experiments on real data, authors only listed several results on some specific edges (not all edges) and showed that COLP is correct on all of these edges. Then naturally I would doubt whether there is cherry-picking, only reporting edges where COLP is correct (and competitors incorrect). Could the authors please report how they choose the edges? And is there any other metric (over all edges/ whole graph, e.g., Table 4) for the real datasets?
 - What does it mean by "out-of-sample prediction (via leave-one-out cross-validation)"? COLP seems to be an unsupervised method. Could the authors please explain more on this?


---
[1] Christopher Meek. ‘Strong Completeness and Faithfulness in Bayesian Networks’. In: Proceedings of the Eleventh Conference on Uncertainty in Artificial Intelligence. UAI’95. Montréal, Qué, Canada: Morgan Kaufmann Publishers Inc., 1995, pp. 411–418

**Limitations:**

Overall, the authors have adequately addressed the limitations: 1) identifiability based on causal sufficiency, 2) extension to multivariate, and 3) more real data. I would expect more elaboration on the hidden assumptions over the data generating process.

**Strengths And Weaknesses:**

+ Pros
  - **Intriguing intuition**: the illustrating example of predicting one's choice of sports based on their height/strength is intuitive: though generally categorical variables do not have natural orderings, they may be ordered in a meaningful way as responses in prediction.
  - **Comprehensive experimental results**: authors assessed the performance of COLP with both synthetic and real-world datasets. On synthetic datasets, three scenarios are considered: small/big number of categories and pairs with hidden confounders. Four well-known real categorical data are evaluated, which shows COLP's better performance than HCR and CE.

+ Cons
  - The **assumption is not well clarified**. The proposed COLP can be viewed as a variation of ordinal-regression-based categorical causal discovery. Then what is exactly the parametric assumption on the data generation process behind COLP and ordinal regression, respectively? This paper does not give a mathematical formulation. See question 1 for details.
  - The theoretical part on **identifiability is poorly explained**, and the basic **examples and results seem wrong (to me)**. See question 2 for details. Please correct me if I did not understand correctly and I'm willing to change my assessment.
  - The **experimental results on real data is broad, but not deep enough**, and is thus not convincing. There is not a quantitive evaluation metric, but only cases studies on some specific edges. Also, there are some small issues about clarity. See question 3 for details.

---

> ### Author Response · Authors · 2022-08-02
> **Response**
>
> We thank the referee for the comments and provide our responses to the major points below.
>
> ### Response to the comment "The identifiability is poorly explained, and the results seem to be wrong".
>
> We believe our results are correct. What's key here is that the link function is fixed and is not a model parameter which would add degrees of freedom to the model. The same idea is commonly used in generalized linear models where the link function is typically fixed rather than estimated nonparametrically. It's worth pointing out that our identifiability theory works for a wide variety of link functions such as logistic and probit (the only requirement is that the derivative is non-zero) so long as they are fixed. If the link function is not fixed and needs to be estimated nonparametrically, then the model becomes much closer to the fully nonparametric multinomial Bayesian networks, which are known to be non-identifiable. This discussion naturally leads to the question that what if the link function is misspecified. For that, we now perform an additional simulation similar to the simulation scenario 1 in our paper with $n=1000$ but a probit link function is used to generate the data. Then we fit the COLP model with the logistic link. The causal identification accuracy is very similar to the non-misspecified case (in fact, there is a slight increase to 0.944). This is also consistent with generalized linear model literature; for example, generally probit regression and logistic regression have very similar classification performance. In summary, as long as the link function is fixed (even if it is misspecified), our model is identifiable.
>
>
> ### Response to the comment "Questions about experimental results on real data".
>
> For the Pittsburgh Bridge and the Abalone data, we consider and report *all* the causal pairs used in the prior work (Cai, et al. 2018). That is, we did not only report the edges that we can identify.
>
> The criterion we used to select causal pairs from the Tubingen and categorical cause-effect data is that at least one of the two variables is a nominal categorical variable, which is the focus of our method.  For example, as a result, only five pairs in Tubingen data satisfy this criterion.
>
> As for the out-of-sample prediction, that refers to the prediction of the effect variable (Y) based on the cause variable (X) using the COLP classifier.
>
> ### Response to the comment "What are the hidden assumptions?"
>
> Our model assumptions are comparable to those made by recent causal discovery methods for categorical data such as Cai, et al. 2018 with, in our opinion, similar degree of testability. The only difference in terms of assumptions is that we assume there is an underlying ordering of the labels of the effect variable whereas Cai, et al. 2018 assumes there is an underlying compact representation of the effect variable. Some earlier work even assumes the discrete data is additive. None of these assumptions is necessarily easier to test/gauge than others. Other assumptions such as causal Markov and causal sufficiency are shared across causal discovery methods for categorical data. While causal insufficient scenarios are relatively well studied for continuous data, they are less explored for categorical data (except for constraint-based methods that do not infer unique causal graphs) because in a way causal discovery for categorical data are inherently harder than continuous data even in a causally sufficient system; for example, nonlinearity or non-Gaussianity, which leads to identifiable continuous causal discovery, does not help here. We will make our assumptions more explicit up front in the paper. The referee mentioned ordinal and multinomial methods. For ordinal method, it is only applicable to ordinal data and for general multinomial methods (such as those based on BDe or BIC), they are only identifiable up to Markov equivalence class.
>
> As for possible extension to multivariate case, we don't think the label permutation is an issue. Consider a simple score-and-search algorithm. To score a graph, we look for all child-parents sets given a graph. For each child-parents set (say, the child is denoted by $Y$ and $X$ are the parents of $Y$), the likelihood of this set is simply given by Equation (1) in the paper. Note that $X$ can be of any dimension because only the label of the response variable (child) is permuted. We can evaluate the likelihood of all the child-parents sets. And the total score is the sum of the log likelihoods. A greedy search is then can be used to search for a good DAG.

---

> > ### Comment · Reviewer_q4Dr · 2022-08-04
> > **Could the authors please explain more about the identifiability?**
> >
> > Thank the authors for your response.
> >
> > "link function is fixed and is not a model parameter that has degrees of freedom": I'm a bit confused - so it's assumed that we already know the **exact parameterized** underlying link function (e.g., logistic) (instead of just assuming there **exists** one), right? According to your analogy to linear models, it's like we assume that we even know the edge weights, instead of just the existence of linear edges.
> >
> > And, could the authors please answer my earlier questions more directly? I was glad to increase my score if your response convinced me. Specifically,
> >
> > + About the example in Figure 1, is the following statement correct or not?
> >   > Seems that the unpermuted $Y$ ($\sigma([1,2,3])=[1,2,3]$) also satisfies a COLP model (with other parameters $\omega, \gamma, \beta$ unchanged): just let $F$ satisfy $F(-1)=0.16, F(0)=0.66, F(1)=0.84, F(2)=0.86$.
> > + Is the permutation-based method (`is_X_to_Y(XY_joint_table)`) to check whether there exists a COLP model underlying the data distribution correct or not?
> >
> > Many thanks.

---

> > > ### Author Response · Authors · 2022-08-04
> > > **Follow-up response**
> > >
> > > We really appreciate the reviewer taking the time to promptly read and respond to our response. We provide our follow-up response below.
> > >
> > > ### Link function
> > > Take the equation below line 106 as an example, which we paste below for convenience. The right-hand side of this equation involves the following quantities: the label permutation $\sigma$, the threshold parameter $\gamma$, the edge weight $\beta$ of $X\to Y$, and the link function $F$. Out of these four quantities, the link $F$ is assumed to be known whereas the other three, which include the edge weight $\beta$, are unknown. If we take a linear model as an analogy, we would use an identity link $F(X)=X$ but the model would be $Y=F(\beta_0+\beta_1X)+\epsilon$ where the regression coefficient still needs to be estimated.
> > >
> > > $ P_{X\to Y}(Y=\ell|X=s)=F(\gamma_{\sigma(\ell)}-X^T\beta)-F(\gamma_{\sigma(\ell)-1}-X^T\beta)$
> > >
> > > ### "About the example in Figure 1" and "The permutation-based method (is_X_to_Y(XY_joint_table))"
> > > If $F$ is not fixed, then we think both points raised by the reviewer regarding the example in Figure 1and using is_X_to_Y(XY_joint_table) to check identifiability would be correct. However, once $F$ is fixed, $F(-1)$, $F(0)$, $F(1)$, $F(2)$ are fixed as well. For example, choosing $F$ to be a probit link (i.e., the CDF of a standard normal distribution),  $F(-1)=0.16$, $F(0)=0.5$, $F(1)=0.84$, $F(0.98)$, which, in fact, gave rise to the example in Figure 1. For the same reason, given a fixed $F$ (i.e., we don't have the degrees of freedom to adjust $F(-1)$, $F(0)$, $F(1)$, $F(2)$), then the existence of COLP model for a given joint probability matrix cannot be checked by the permutation-based method (is_X_to_Y(XY_joint_table)). To empirically see this, we now used command "joint = np.random.dirichlet(np.ones(9), size=1).reshape((3, 3))" to randomly generate a 3 by 3 probability matrix, and then pass it to "is_X_to_Y(joint)". For example, "is_X_to_Y(joint)" returns TRUE for the following joint probability matrix,
> > >
> > > |        |            |            |            |
> > > |--------|------------|------------|------------|
> > > | P(Y,X) | X=1        | X=2        | X=3        |
> > > | Y=1    | 0.00266916 | 0.08399492 | 0.2562149  |
> > > | Y=2    | 0.11548289 | 0.10143557 | 0.05521942 |
> > > | Y=3    | 0.23619425 | 0.05821573 | 0.09057317 |
> > >
> > > Using this probability matrix, we generated 100,000 observations and regressed Y on X using COLP. The estimated conditional probability $P(Y|X)$ is given by,
> > >
> > > |        |      |      |      |
> > > |--------|------|------|------|
> > > | P(Y\|X) | X=1  | X=2  | X=3  |
> > > | Y=1    | 0.07 | 0.38 | 0.57 |
> > > | Y=2    | 0.21 | 0.34 | 0.29 |
> > > | Y=3    | 0.71 | 0.28 | 0.14 |
> > >
> > > whereas the true conditional probability is given by,
> > >
> > > |        |      |      |      |
> > > |--------|------|------|------|
> > > | P(Y\|X) | X=1  | X=2  | X=3  |
> > > | Y=1    | 0.01 | 0.34 | 0.64 |
> > > | Y=2    | 0.33 | 0.42 | 0.14 |
> > > | Y=3    | 0.67 | 0.24 | 0.23 |
> > >
> > > Even with 100,000 observations, there is still a significant bias; compare e.g., the second rows P(Y=2|X). By contrast, for the example in Figure 1, the estimate of the $P(Y|X)$ gets arbitrarily close to the truth quickly as sample size increases. This toy simulation suggests that is_X_to_Y(XY_joint_table) cannot be used when the link function $F$ is fixed.

---

> > ### Comment · Reviewer_q4Dr · 2022-08-09
> > **RE Response**
> >
> > Thank the authors for your detailed response. My concerns about the correctness of this paper is well addressed.
> > As promised I will increase my score to 5. However I could not give any higher, because the assumption that "not only a link function F exists but we also know it exactly" is a bit too restrictive to me. Any try to relax it would be great.

---

### Official Review · Reviewer_WFBs · 2022-07-10

**Rating:** 5
**Confidence:** 4
**Soundness:** 4 excellent
**Presentation:** 3 good
**Contribution:** 4 excellent

**Summary:**

The authors propose a novel method to do pairwise orientation (and adjacency discovery?) for categorical variables with > 2 categories.

**Questions:**

* Could you include MEC methods in your analyses as well? This might require using more variables and selecting good methods, but it is possible.

* Could the situation for identifiability with binary variables be made clear?

* Could it be clarified whether COLP is able to detect the nonexistence of pairwise edges and is able to conclude "undirected" if in fact a direction should not be inferred?


**Limitations:**

I did not see societal impact addressed; nor do I see it as much of an issue for this method.

**Strengths And Weaknesses:**

Strengths:

* The idea is based on a sound principle.

* The creation of a categorical pairwise causal dataset is a great idea. Hopefully others can try their hand at it.

* The theory given for the COLP algorithm intuitively makes sense, as does the algorithm provided.

* I wondered whether the "natural orderings" found by COLP corresponded to intuitively natural ordering, but there were comments int he experimental section to this effect, so thanks.

Weaknesses:

* A main weakness of this paper is that is simply discounts out of hand the Markov Equivalence Class (MEC) approach to causal discovery. In fact, it is not even mentioned in the paper so far as I can see, despite the fact that it has been a major methodology for many years now and is still actively studied.

Out of curiosity, I loaded up one of the the Pittsburgh Bridges dataset studied here removed the first column, and ran the novel GRaSP algorithm (UAI 2022) on it with the BDeu score with _all 12 variables_, and two of the ground truth edges identified by the authors were correctly oriented, with paths existing for the other ground truth edges. (It seems the authors only studied 4 of the 12 variables, suggesting that a latent variable algorithm may have been necessary.) So that strategy should not be discounted.

There is in fact a strategy in literature of using an MEC strategy to find and orient as many edges as possible and then to use a pairwise strategy to orient the remaining edges. Again, this should not be discounted, and "causal discovery" should not be taken to mean just pairwise (or "bivariate"--author's term) causal discovery.

There may be a question as to how the various methods (MEC methods included) do at detecting the directionality of the various edges. This is a sensitivity question and can be directly addressed.

I'm also very familiar with the Abalone dataset, which is mixed continuous/discrete. Again, loading it up and running the GRasP algorithm on it with the mixed Conditional Gaussian score renders SEX as exogenous with respect to all of the other variables, which it should be, as female abalone are larger in size systematically than male abalone. Again, this is an MEC approach.

* It was gratifying to see COLP results for even some of the Tubingen pairwise examples as well, though #54 is of course an 8-variable mixed continuous/discrete example ("auto-mpg") from the UCI Repository that can be handled using MEC method; I tried GRaSP on it as well with the Conditional Gaussian score, and it gets it right.

* In Section 3.1, two competing categorical discovery methods are mentioned, HCR and CE; why are these not mentioned in the introduction so help situate the COLP algorithm immediately in the literature?

* The word "identifiable" needs to be clarified here, as the MEC approach would take X--Y to be identifiable under certain assumptions, as meaning that there is either an edge X->Y or and edge X<-Y. It should be particularly mentioned, I think, that identifiability in this context means correctly identifying which of X->Y or X<-Y is correct.

* Also, it is not clear to me based on what is said what identifiability extends to the binary case; if not, this should be stated up front.

* I do wonder if the COLP method will correctly infer that an edge cannot be oriented pairwise if in fact it cannot, or if it will tend to guess one direction or the other?

* In Figure 2, I conclude that COLP should not be attempted with sample sizes less than 400, which is good to know. I find it interesting that 5 is considered a "small" number of categories--this is not the world I live in.

---

> ### Author Response · Authors · 2022-08-02
> **Response**
>
> We thank the referee for the comments and provide our responses to the major points below.
>
> ## Question 2 and Weakness 5
> The proposed model is not identifiable for binary variable as the permutation would not make a difference (e.g., a logistic regression would be the same no matter how the response variable is labeled). We will make it clear in our paper.
>
> ## Question 3
> Due to Theorems 1 and 2, the proposed method, as it is, cannot detect "undirected" edges because the likelihoods of any two competing causal models are almost surely not equal. However, if desired, one can set up a threshold and only declare causal identification if the likelihood ratio is larger than the threshold and otherwise declare it to be an undirected edge. Such threshold may be chosen based on the asymptotic distribution of the likelihood ratio if it can derived. As for the identification of "non-existence" of an edge, yes, our identifiability theories allow that.
>
> ## Weakness 1 and Question 1
> MEC-based methods for discrete data are now added to our data analyses. Specifically, we consider four methods: (1) the multinomial Bayesian networks with BIC and (2) BDe scores, (3) the PC algorithm with mutual information-based conditional independence test, and (4) suggested GRaSP.
>
> For the Bridge and Abalone data, we are able to reproduce the referee's finding using GRaSP. The results for all the methods are summarized below where "missing" means no edge is found, "correct" means the direction is correctly identified, "wrong direction" means an directed edge is found but the direction is wrong, "path" means there is a path between the two non-adjacent variables, and "undirected" means an edge is found but the direction cannot be determined.
>
> ### Bridge
> | Methods           | BDe             | PC  | BIC     | GRaSP    |
> |-------------------|-----------------|---------------------|---------|----------|
> | Erected -> Span   | missing         | path                | missing | path     |
> | Material -> Span  | correct         | wrong direction     | correct | correct  |
> | Material -> Lanes | missing         | path                | missing | path     |
> | Purpose -> Type   | wrong direction | path                | missing | correct  |
>
>
> ### Abalone
> | Methods         | BDe | PC  | BIC | GRaSP       |
> |-----------------|-------------------|---------------------|-------------------|-------------|
> | Sex -> Length   | missing           | missing             | missing           | path        |
> | Sex -> Diameter | missing           | missing             | missing           | undirected  |
> | Sex -> Height   | missing           | missing             | missing           | correct     |
>
>
> We also apply these four methods to the categorical cause-effect pairs data we created. The results are summarized below where we further divide path into directed path, undirected path, and directed path with wrong direction (called reversed directed path). Note that we do not apply MEC-based methods to datasets with only two variables.

---

> > ### Author Response · Authors · 2022-08-02
> > **Response (continued)**
> >
> > We also apply these four methods to the categorical cause-effect pairs data we created. The results are summarized below where we further divide path into directed path, undirected path, and directed path with wrong direction (called reversed directed path). Note that we do not apply MEC-based methods to datasets with only two variables.
> >
> > ### Categorical Cause-Effect Pairs
> > | Data      | Cause  | Effect      | BDe                    | PC         | BIC                    | GRaSP (BDeu)     |
> > |-----------|--------|-------------|------------------------|------------|------------------------|------------------|
> > | painters  | School | Composition | missing                | missing    | missing                | missing          |
> > | painters  | School | Drawing     | missing                | missing    | missing                | missing          |
> > | painters  | School | Colour      | missing                | missing    | missing                | missing          |
> > | painters  | School | Expression  | missing                | missing    | missing                | missing          |
> > | birthwt   | race   | low         | missing                | missing    | missing                | undirected       |
> > | bacteria  | trt    | y           | missing                | missing    | reversed directed path | missing          |
> > | survey    | Sex    | Clap        | missing                | missing    | missing                | missing          |
> > | survey    | Sex    | Fold        | missing                | missing    | missing                | missing          |
> > | oats      | B      | Y           | missing                | missing    | missing                | missing          |
> > | oats      | V      | Y           | missing                | missing    | missing                | missing          |
> > | oats      | N      | Y           | missing                | undirected | missing                | missing          |
> > | crabs     | sp*sex | FL          | missing                | missing    | missing                | missing          |
> > | crabs     | sp*sex | RW          | directed path          | undirected | missing                | correct          |
> > | crabs     | sp*sex | CL          | correct                | missing    | missing                | missing          |
> > | crabs     | sp*sex | CW          | directed path          | missing    | missing                | missing          |
> > | crabs     | sp*sex | BD          | directed path          | missing    | missing                | missing          |
> > | fgl       | type   | RI          | directed path          | missing    | missing                | undirected path  |
> > | immer     | Var    | Y1          | correct                | missing    | missing                | missing          |
> > | immer     | Var    | Y2          | missing                | missing    | missing                | missing          |
> > | immer     | Loc    | Y1          | correct                | missing    | missing                | undirected       |
> > | immer     | Loc    | Y2          | missing                | missing    | missing                | missing          |
> > | minn38    | sex    | phs         | wrong direction        | undirected | wrong direction        | undirected       |
> > | minn38    | fol    | hs          | reversed directed path | undirected | reversed directed path | undirected path  |
> > | minn38    | fol    | phs         | wrong direction        | undirected | wrong direction        | undirected       |
> > | ﻿UScereal | mfr    | ﻿shelf      | missing                | missing    | missing                | undirected path  |
> > | ﻿UScereal | mfr    | vitamins    | missing                | missing    | missing                | missing          |
> >
> > In the paper, we will introduce MEC-based methods and note that based on the results above, MEC-based methods can identify the connection (which may be a path, directed edge, or an undirected edge) between two categorical variables under certain scenarios.

---

> > > ### Comment · Reviewer_WFBs · 2022-08-09
> > > **Encouraging!**
> > >
> > > This is encouraging, thanks! Ground for increasing the score. I'll increase to 5.

---

### Official Review · Reviewer_ettZ · 2022-07-10

**Rating:** 5
**Confidence:** 5
**Soundness:** 4 excellent
**Presentation:** 3 good
**Contribution:** 2 fair

**Summary:**

This paper proposed classification with the optimal label permutation (COLP) method for causal discovery from categorical data. The main idea is that the ordinal regression is identifiable given the correct permutation.

**Questions:**

See the weaknesses above.

**Limitations:**

No potential negative societal impact.

**Strengths And Weaknesses:**

Strengths:
- The proposed COLP is able to discover the causal relationship from categorical data.
- It shows that COLP is identifiable almost surely.

Weaknesses:
- This work can be seen as a direct increment of [1], and the novelty is somewhat limited.
- This work heavy rely on the label permutation on ordinal regression, and thus it is necessary to conduct an ablation study that remove the label permutation to show the supremacy of COLP.
- The proposed greedy algorithm for learning permutation seems inefficacy under the large categories, and it should show how the number of categories affects the performance and whether the learned permutation is correct.
[1] Ni Y, Mallick B. Ordinal Causal Discovery[C]//The 38th Conference on Uncertainty in Artificial Intelligence. 2022.

---

> ### Author Response · Authors · 2022-08-02
> **Response**
>
> We thank the referee for the comments and provide our responses below.
> ### Weakness 1
> Although the proposed COLP is an extension of the recent causal discovery method (Ni and Mallick, 2022) for ordinal data, it has four novel contributions:
> - COLP can be applied to any categorical data whereas Ni and Mallick, 2022 is only applicable to ordinal data, and COLP strictly contains Ni and Mallick, 2022 as a special case by fixing the label permutation.
> - Because of the label permutation, our identifiability theory is new and is more complex than Ni and Mallick, 2022.
> - We in addition provide an asymptotic guarantee of the proposed causal discovery method (Theorem 2), which is not provided in Ni and Mallick, 2022.
> - We have created a categorical causal discovery benchmark dataset.
>
> ### Weakness 2
> We follow the reviewer's suggestion to consider models without label permutation in simulations. Specifically, we fix the permutation at different label orderings. We test this method in Simulation Scenario 1 with $n=1,000$. The results are presented below where Kendall's Tau quantifies the correlation between the fixed label permutation and the true label permutation. As expected, the causal identification accuracy increases as the Kendall's Tau approaches 1. The performance of the COLP method with unknown permutation is close to Kendall's Tau=0.8. This ablation study stresses the importance of having label permutation as a parameter because otherwise the inference can be very wrong if the permutation is fixed to a bad ordering.
> | Kendall's Tau | 0    | 0.2  | 0.4  | 0.6  | 0.8  | 1  |
> |---------------|------|------|------|------|------|----|
> | Accuracy      | 0.17 | 0.33 | 0.61 | 0.82 | 0.96 | 1  |
>
> ### Weakness 3
> We follow the reviewer's suggestion to investigate how estimation of causal direction and label permutation vary as the number of categories increases. Specifically, in Simulation Scenario 1 with $n=1,000$, we consider 10 different numbers of categories from 3 to 12. The results are reported below where the second row is the accuracy of causal identification and the third row is the Kendall's Tau measuring the correlation between the estimated and true label permutations. For both metrics, a value close to 1 indicates good performance. We find that the performance of COLP is relatively stable with respect to the number of categories, especially for causal identification, which is the main focus of the paper.
> | # of Categories             | 3    | 4    | 5    | 6    | 7    | 8    | 9    | 10   | 11   | 12    |
> |---------------|------|------|------|------|------|------|------|------|------|-------|
> | Accuracy      | 0.86 | 0.89 | 0.91 | 0.91 | 0.88 | 0.91 | 0.9  | 0.9  | 0.9  | 0.88  |
> | Kendall's Tau | 0.94 | 0.92 | 0.93 | 0.92 | 0.88 | 0.9  | 0.88 | 0.87 | 0.87 | 0.84  |

---

### Official Review · Reviewer_Q1zD · 2022-07-11

**Rating:** 5
**Confidence:** 3
**Soundness:** 3 good
**Presentation:** 3 good
**Contribution:** 2 fair

**Summary:**

The authors propose a method for cause-effect inference when the causal pair consists of two categorical variables. Essentially, the authors fit an ordinal regression model and see in which direction they can fit a more parsimonious model. The parsimony is motivated by the assumption that certain orderings of the input data fit better with the variables in the outcome data, even when there is no natural ordering in the data. That way, the authors can fit a so-called *COLP (classification with optimal label permutation)* model in which they search for the permutation of the inputs that leads to the best maximum likelihood fit.

**Questions:**

Which anti-causal model do you choose in Figure 1, if it is not a COLP-based causal model?

In Figure 2, have you considered adding some horizontal jitter so that COLP-Exhaustive is more visible (in Scenarios 1 and 3)? Since the accuracy is very similar, it is not easy to see that both are plotted together.

In Figure 3, why would error bars hinder clarity? It seems that the improvement in Kendall's Tau is significant enough to be visible. Why not make three figures, one for each scenario, since there is enough space available?

In Subsection 3.1.3, what is the strength of the confounder? I appreciate that the authors show robustness to violations of causal sufficiency, but I am rather surprised that the method performs very closely to how it performs in Scenario 1.

In Subsection 3.2.1, what does RR stand for? Rest and relaxation? Other variables could probably be explained in more detailed as well, but $\hat{\beta}_{RR}$ is mentioned to show that the ordering is consistent, so the acronym should be explained.

**Limitations:**

The authors do not discuss any potential negative societal impacts of their work, but it is difficult to foresee any for this particular task.

**Strengths And Weaknesses:**

The method proposed is straightforward and appears sound. The experimental section is comprehensive and contains experiments on both simulated and real-world data. The results are convincing and suggest that the COLP-greedy approach performs sufficiently well for the chosen experiments relative to the computationally-prohibitive COLP-exhaustive method.

In terms of weaknesses, the COLP method relies on the assumption that some particular ordering in the categories makes more sense than other orderings. It is hard to gauge how frequently this assumption will hold in practice. Furthermore, one must assume no hidden confounders, which is an important limitation that the authors admit. The clarity of presentation could be improved (see below).

Detailed comments:
- p1 (page 1), l27 (line 27): "the $Y$ admit" $\to$ '$Y$ admits'
- p1, l31: I think you mean 'or' instead of "and".
- p2, l39: "... natural orderings. But ..." $\to$ '... natural orderings **, but** ...'
- p2, l68: Why include $\gamma_L$, unlike $\gamma_0$, in the list of thresholds from line 65, if $\gamma_L = \infty$? Later on, it is no longer included (p3, l109).
- p2, l75: At the end of the sentence there is a dangling left parenthesis.
- p3, l85: "contraint" $\to$ 'constraint'
- p3, l108: Add 'the' before "causal model".
- p3, l111 & l117: "casual" $\to$ 'causal'
- p3, l132: Add 'a' after "such".
- p3, l138: "does" $\to$ 'do'
- p3, l141: "confounder" $\to$ 'confounders'
- p6, Figure 2 caption: "COLP-Exhausitve" $\to$ 'COLP-Exhaustive'
- p8, l233: "pair" $\to$ 'pairs'

---

> ### Author Response · Authors · 2022-08-02
> **Response**
>
> We thank the referee for the comments and provide our responses below.
>
> ### Weakness 1
> Our model assumptions are comparable to those made by recent causal discovery methods for categorical data such as Cai, et al. 2018 with, in our opinion, similar degree of testability. The only difference in terms of assumptions is that we assume there is an underlying ordering of the labels of the effect variable whereas Cai, et al. 2018 assumes there is an underlying compact representation of the effect variable. Some earlier work even assumes the discrete data is additive. None of these assumptions is necessarily easier to test/gauge than others. Other assumptions such as causal Markov and causal sufficiency are shared across causal discovery methods for categorical data. While causal insufficient scenarios are relatively well studied for continuous data, they are less explored for categorical data (except for constraint-based methods that do not infer unique causal graphs) because in a way causal discovery for categorical data are inherently harder than continuous data even in a causally sufficient system; for example, nonlinearity or non-Gaussianity, which leads to identifiable continuous causal discovery, does not help here. We will make our assumptions more explicit up front in the paper.
>
> ### Question 1
> The anti-causal model would be a saturated/nonparametric multinomial/categorical casual model that cannot be represented by a parsimonious/parametric COLP model.
>
> ### Question 2
> We will add jitter to Figure 2 for clarity.
>
> ### Question 3
> We will make separate plots with error bars.
>
> ### Question 4
> The strength of the confounder is chosen to be the same as the causal effect between the observed variable. We are glad to see that our method performs quite well in the presence of moderate confounding effects.
>
> ### Question 5
> RR stands for Rail Road. We will note it in the paper.

---

> > ### Comment · Reviewer_Q1zD · 2022-08-09
> > **Thanks**
> >
> > Thank you the response and for addressing my concerns.

---

### Meta-Review · Area_Chair_qvMv · 2022-08-28

**Recommendation:** Accept
**Confidence:** Certain

**Metareview:**

All reviewers are convinced by the scientific soundness and evaluation results about this paper.
The reviewers had some concerns regarding clarity and evaluation but in general liked
various aspects of the paper. The authors did a good job of addressing the reviewers' concerns
so acceptance is recommended.


**Award:**

No

---

### Decision · Program_Chairs · 2022-09-14

Accept